# New Epigenetic Modifier Inhibitors Enhance Microspore Embryogenesis in Bread Wheat

**DOI:** 10.3390/plants13060772

**Published:** 2024-03-08

**Authors:** Isabel Valero-Rubira, María Pilar Vallés, Begoña Echávarri, Patricia Fustero, María Asunción Costar, Ana María Castillo

**Affiliations:** Department of Genetics and Plant Breeding, Aula Dei Experimental Station, Spanish National Research Council (EEAD-CSIC), 50059 Zaragoza, Spain; ivalero@eead.csic.es (I.V.-R.); valles@eead.csic.es (M.P.V.); echavarri@eead.csic.es (B.E.); patriciafustero@eead.csic.es (P.F.); acostar@eead.csic.es (M.A.C.)

**Keywords:** bread wheat, microspore embryogenesis, doubled haploid, epigenetic modifiers, aurora kinase inhibitor II, spontaneous chromosome doubling

## Abstract

The use of doubled haploid (DH) technology enables the development of new varieties of plants in less time than traditional breeding methods. In microspore embryogenesis (ME), stress treatment triggers microspores towards an embryogenic pathway, resulting in the production of DH plants. Epigenetic modifiers have been successfully used to increase ME efficiency in a number of crops. In wheat, only the histone deacetylase inhibitor trichostatin A (TSA) has been shown to be effective. In this study, inhibitors of epigenetic modifiers acting on histone methylation (chaetocin and CARM1 inhibitor) and histone phosphorylation (aurora kinase inhibitor II (AUKI-II) and hesperadin) were screened to determine their potential in ME induction in high- and mid-low-responding cultivars. The use of chaetocin and AUKI-II resulted in a higher percentage of embryogenic structures than controls in both cultivars, but only AUKI-II was superior to TSA. In order to evaluate the potential of AUKI-II in terms of increasing the number of green DH plants, short and long application strategies were tested during the mannitol stress treatment. The application of 0.8 µM AUKI-II during a long stress treatment resulted in a higher percentage of chromosome doubling compared to control DMSO in both cultivars. This concentration produced 33% more green DH plants than the control in the mid-low-responding cultivar, but did not affect the final ME efficiency in a high-responding cultivar. This study has identified new epigenetic modifiers whose use could be promising for increasing the efficiency of other systems that require cellular reprogramming.

## 1. Introduction

The effects of climate change and the growth of the world’s population, which is estimated to reach 9–10 billion by 2050, have created an urgent need to develop new crop varieties with high productivity and improved adaptability to evolving conditions. This is especially relevant for bread wheat (*Triticum aestivum* L.), which is a primary protein source in the human diet, accounting for approximately 20% of the daily calorie intake. Therefore, it is estimated that wheat production will be required to rise by more than 50% by 2050 [1]. In this context, doubled haploid (DH) production technology is highly relevant, since completely homozygous plants can be obtained within one generation. Furthermore, the use of DH lines enhances selection efficiency and facilitates the development of new varieties within 4–6 years, thereby reducing costs compared to traditional breeding methods [2].

Microspore embryogenesis (ME) displays the greatest potential for DH plant production. In ME, stress treatment triggers microspores to shift towards an embryogenic pathway, which is followed by the growth of haploid and DH embryos and plants when cultured in vitro [3]. However, the efficiency of wheat ME is highly dependent on genotype [4,5], which limits the practical application of this methodology to plant breeding programs. The low number of microspores that can effectively change their developmental fate, the similarly low number of high-quality embryos, and the high percentage of albino plantlets are primary limiting factors [6].

Epigenetic mechanisms regulate key processes during ME induction, including response to stress treatment, apoptosis and cell death, and symmetric and asymmetric division [7,8,9]. In this sense, the application of epigenetic modifiers, such as histone deacetylase inhibitors (trichostatin A (TSA), sodium butyrate (NaB), suberoylanidide hydroxamic acid (SAHA), histone lysine methyltransferase G9a inhibitor (BIX-01294), and DNA methyltransferase inhibitor 5-azacitydine (AzaC), has effectively enhanced ME in several species [10,11,12,13,14,15,16,17,18].

For wheat ME, various epigenetic modifiers have been tested on the basis of results from other species. For instance, the application of TSA during or after stress treatment or in a culture medium increases the number of green plants [19,20,21]. Conversely, the application of NaB and BIX-01294 in a culture medium results in fewer green plants [20]. Our group recently obtained new information on changes in gene expression during ME induction in wheat. This data enabled us the identification of histone methylation and phosphorylation mechanisms involved in ME, as well as the selection of candidate compounds that act on them.

Histone can be methylated at lysine and arginine residues, with methylation at lysine residues of histone H3 and H4 being the most observed [22]. Lysine methylation is mediated by histone lysine (K) methyltransferases (HKMTs). Coactivator-associated arginine methyltransferase 1 (CARM1) is a member of the arginine methyltransferase (PRMT) family that specifically acts on histones and other chromatin-associated proteins. Both HKMTs and PMRTs play key roles in plant growth, development, and stress response [23,24]. Chaetocin is a thiodiketopyrazine that acts as an inhibitor of histone lysin methyltransferase SU(VAR) 3-9 [25], and the CARM1 inhibitor specifically influences CARM1.

Plant aurora kinases (AUKs), phosphorylate histone H3 at serine 10, as well as other proteins. These kinases are key regulators of mitosis and meiosis [26]. In *Arabidopsis*, aurora kinases are classified as follows: α-aurora and β-aurora [27]. Aurora kinase inhibitor II (AUKI-II) is a synthetic molecule that inhibits α-aurora kinases [28], and hesperadin restricts β-aurora kinases [29,30].

This study compares the effectiveness of chaetocin, CARM1 inhibitor, AUKI-II, and hesperadin with that of TSA in terms of inducing ME in two cultivars of bread wheat with different embryogenic responses. The epigenetic modifiers demonstrated differences in the percentage of embryogenic structures after 10 days in culture, with AUKI-II showing the highest potential. Therefore, we studied the effect of the application of AUKI-II, in combination with short- and long-term mannitol stress treatment, on the final efficiency of ME.

## 2. Results

### 2.1. Selection of Candidate Epigenetic Modifiers

Our group conducted a RNA-seq analysis of microspores during induction and early stages of culture in wheat. The analysis revealed that the stress treatment caused the down-regulation of genes involved in histone methylation and histone phosphorylation. These genes included histone-lysine and histone-arginine N-methyltransferases, as well as aurora kinase (Appendix A). Commercial compounds that act by inhibiting the enzymes encoded by these genes were selected as putative inducers of ME. 

### 2.2. Screening of Different Epigenetic Modifiers to Induce ME in Wheat 

During in vivo pollen development, uninucleate microspores contain large central vacuoles, with the nuclei positioned close to the microspore cell wall (Figure 1A(a,b)). Microspores at the late uninucleate stage undergo asymmetric division, resulting in the formation of a vegetative and a generative cell (Figure 1A(c,d)). Subsequently, the generative cell undergoes a second pollen division, producing two sperm cells (see Figure 1A(e)).

This study evaluates the potential of chaetocin, a CARM1 inhibitor, AUKI-II, and hesperadin to induce ME by quantifying the different types of structures present after 2, 4, and 10 days in culture. TSA was included as an efficient epigenetic modifier to induce ME. All compounds were applied at concentrations of 0.4 µM during a 24 h stress treatment period in the cultivars Pavon and Caramba. A control (SM liquid medium, L-CM) and control DMSO (SM liquid medium with 1% DMSO final concentration in the medium, L-CM+ DMSO) were also included.

Most representative structures observed at the early stages of culture after DAPI staining are presented in Figure 1B. Uninucleate microspores showed large and decondensed nuclei (Figure 1B(a)). We observed bicellular structures resulting from symmetric division with nuclei of the same size and condensation (Figure 1B(b)), and pollen-like bicellular structures with small generative-like cells with condensed nuclei and vegetative-like cells with large, decondensed nuclei (Figure 1B(c)). Different types of tricellular structures were identified depending on the outcome of the first division and the participating nucleus in the subsequent division. These included a pollen-like structure showing the vegetative-like cell and the two sperm-like cells (Figure 1B(d)), a vegetative structure with two vegetative-like cells and one generative-like cell (Figure 1B(e)), and a structure with three cells containing nuclei of similar sizes (Figure 1B(f)). The last two structures exhibited a deviation from the developmental pattern towards pollen, indicating that they had already been reprogrammed and therefore could be considered tricellular embryogenic structures. We also identified more advanced structures in development, with four cells (tetracellular) of similar sizes (Figure 1B(g)). Further proliferation produced larger embryogenic structures enclosed inside the exine (multicellular structures) (Figure 1B(h,i)). After 10 days in culture, several multicellular structures broke down the exine, allowing for the formation of pro-embryos (Figure 1B(j)).

For both cultivars, and for all epigenetic modifiers and controls, microspores and bicellular structures were mostly observed after two days in culture (2dC) (Figure 2, Appendix A). In Pavon, TSA (L-0.4TSA) showed the highest percentage of bicellular structures (68.9%), while the control DMSO yielded the lowest (53.6%). In Caramba, the percentages of bicellular structures ranged from 75.0% with TSA to 64.0% with hesperadin (L-0.4Hesperadin). We observed a small proportion of tricellular structures (lower than 5%) and an even lower proportion of tetracellular structures (lower than 1%) across all treatments in both cultivars.

Following four days in culture (4dC), there was a slight increase in the percentage of tricellular and tetracellular embryogenic structures with most of the epigenetic modifiers in both cultivars (Figure 2, Appendix A). In Pavon, hesperadin yielded the highest percentage of tricellular pollen-like structures (6.0%), and AUKI-II (L-0.4AUKI-II) produced the highest percentage of tricellular embryogenic structures (6.1%) (Figure 2A, Appendix A). No significant differences were found between treatments in terms of the percentage of tetracellular structures. However, hesperadin yielded the highest percentage (2.0%), while the control DMSO yielded the lowest (0.0%). In Caramba, both TSA (16.1%) and CARM1 inhibitor (L-0.4CARM1I) (13.7%) resulted in higher percentages of pollen-like structures than the control (4.5%) and the other epigenetic modifiers. These two compounds also yielded the lowest percentage of tricellular embryogenic structures, with 0.0% (Appendix A, Appendix A). Similar percentages of tricellular and tetracellular embryogenic structures were observed in all treatments. However, AUKI-II and TSA showed the highest values of tricellular and tetracellular embryogenic structures (3.0% and 1.2%, respectively). Multicellular structures were also observed at 4dC in both cultivars when exclusively using TSA, and also with AUKI-II in Pavon (Appendix A).

After 10 days in culture (10dC), the percentages of tricellular pollen-like structures were lower than or similar to those observed at 4dC, with the exception of hesperadin in Caramba. However, the number of tricellular, tetracellular, and multicellular embryogenic structures increased at different rates depending on the compound and the cultivar (Figure 2, Appendix A, Appendix A). In Pavon, AUKI-II and TSA produced the highest percentage of tricellular embryogenic structures (approximately 11.7%) (Figure 2A, Appendix A). The CARM1 inhibitor produced a higher percentage of tetracellular structures (10.3%) than the control (2.3%). All the epigenetic modifiers, with the exception of hesperadin, resulted in higher percentages of multicellular structures than those observed with the control. AUKI-II produced the highest percentage (21.8%), which was slightly higher than TSA (18.1%). Both percentages were almost doubled compared to the control with DMSO (9.8%).

In Caramba, chaetocin produced the highest percentage of tricellular embryogenic structures (19.8%), a rate slightly higher than that of the control (14.6%) (Figure 2B, Appendix A). Additionally, chaetocin and AUKI-II resulted in a four-fold increase in the percentage of tetracellular structures (approximately 8.0%) compared to the control (1.8%). However, AUKI-II and TSA produced the highest percentages of multicellular structures (5.6% and 4.7%, respectively). AUKI-II doubled this percentage compared to the control, but this was not significant. Remarkably, the CARM1 inhibitor did not produce multicellular structures.

As expected, all treatments resulted in a lower percentage of multicellular structures in Caramba than in Pavon. AUKI-II, TSA, and control DMSO produced the structures with the highest numbers of cells in both cultivars (Appendix A).

### 2.3. Effect of Aurora Kinase Inhibitor II on the Efficiency of Green DH Plant Production

Following the screening of epigenetic modifiers, the effect of applying different concentrations of AUKI-II during short and a long mannitol stress treatments was studied. The final ME efficiency was evaluated by recording the number of pro-embryos (PEMB: multicellular structure outside the exine), embryos (EMB: embryos with a well-developed embryo axis), green plants (GP), albino plants (AP), and spontaneously green DH plants (GPDH).

#### 2.3.1. Application of Aurora Kinase Inhibitor II during a 24 h Stress Treatment

The efficacy of 24 h of treatment with 0.4 and 0.8 µM AUKI-II (L-0.4AUKI-II and L-0.8AUKI-II, respectively) in SM liquid medium was evaluated in relation to the high-responding cultivar Pavon. We also included control cultures in SM liquid medium (L-CM) and control DMSO in SM liquid medium plus 0.1% DMSO (L-CM + DMSO). The application of a 0.4 µM TSA in SM liquid medium was also utilized as an efficient epigenetic modifier to increase ME (L0.4TSA).

Analysis of variance (ANOVA) showed no significant differences among the two controls and treatments with AUKI-II or TSA for any of the studied variables, except for the percentages of regeneration and spontaneous doubling. Means separation is presented in Table 1. The DMSO significantly enhanced the percentage of regeneration (47.3%) and reduced the percentage of spontaneous doubling (46.4%) compared to the control (29.0% and 65.8%, respectively). The TSA and control DMSO showed the lowest value of spontaneous doubling (approximately 46.0%). However, a significantly higher percentage of doubling was obtained with 0.4 and 0.8 µM AUKI-II (59.1% and 68.5%, respectively) compared to the control DMSO, resulting in similar values to that of the control. The highest final ME efficiency (number of green DH plants) was achieved with 0.8 µM AUKI-II (15.2), although there were no significant differences with respect to either the control or, indeed, TSA.

#### 2.3.2. Application of Aurora Kinase Inhibitor II during a 5-Day Stress Treatment

The effectiveness of the application of 0.4, 0.8, and 1.2 µM AUKI-II (0.4AUKI-II, 0.8AUKI-II and 1.2 AUKI-II, respectively) during a 5-day stress treatment in SM medium was evaluated on the Pavon and Caramba cultivars. Control cultures in SM medium (CM) and control DMSO in SM medium with 1% DMSO (CM + DMSO) were also included for evaluation. 

ANOVA showed significant differences between the cultivars for all the variables except for the number of albino plants and the percentage of regeneration (Table 2). For instance, in Pavon, the control produced approximately twice the number of pro-embryos, embryos, and percentage of green plants, and five times the number of green plants and green DH plants compared to Caramba (Table 3, Figure 3). No statistically significant differences were found between the treatments, nor a genotype X treatment interaction for any of the variables (Table 2). However, there was a trend towards significance in the interaction for the number of pro-embryos (*p*-value of 0.097).

Mean separation showed that the number of pro-embryos significantly decreased when treated with 0.4 µM and 1.2 µM AUKI-II (647.3 and 603.0, respectively) compared to the control (767.0) in Pavon, whereas no significant differences among treatments were observed in Caramba (Table 3). Neither the control DMSO nor any of the concentrations of AUKI-II had a noticeable effect on the number of embryos developed (Figure 3A–E) or the percentage of green plants (Figure 4) compared to control in Pavon. However, the highest concentration of AUKI-II (1.2 µM) increased the percentage of regeneration as compared to lower concentrations and control DMSO, although its value was similar to that of the control (Table 3). In Caramba, slightly higher numbers of embryos and green plants were produced, with 1.2 µM and 0.8 µM AUKI-II, respectively, compared to the lower concentration of AUKI-II and both controls (Table 3, Figure 3F–J).

Comparing the data from Pavon after short and long treatments, the long treatment resulted in approximately twice the number of green DH plants (Table 1 and Table 3).

The effect of AUKI-II treatments on the percentage of spontaneous doubling and the number of green DH plants also depended on the cultivar (Table 3). In the mid-low-responding cultivar Caramba, the control DMSO and 0.4 µM AUKI-II treatments caused significant decreases in the percentage of spontaneous doubling (44.3% and 39.0%, respectively) compared to the control (61.7%). By contrast, 0.8 µM and 1.2 µM AUKI-II significantly increased this variable (63.1% and 58.0%) compared to the control DMSO, resulting in values similar to those of the control (61.7%). Concerning the final ME efficiency, 0.8 µM AUKI-II significantly increased the number of green DH plants in this cultivar compared to the control DMSO (6.1 and 3.1, respectively). Although no significant differences between 0.8 µM AUKI-II and the control were obtained for this variable, this concentration increased its value by 33%. However, in the high-responding cultivar Pavon, only a slight increase in the percentage of chromosome doubling was observed with 0.8 µM AUKI-II (64.5%) versus the control DMSO and the control (56.0% and 57.3%, respectively).

## 3. Discussion

Effective strategies for the enhancement of ME are based on the application of epigenetic modifiers during or after stress treatment or in culture medium for short or long periods. In wheat, only the histone deacetylase inhibitor TSA has been found to be effective [19,20,21], with the success rates varying among genotypes. However, there are still many genotypes that are unresponsive to ME [5].

In order to broaden the range of strategies available to induce ME in non-responsive wheat genotypes, it is necessary to test other epigenetic modifiers that have been effective in other systems but whose use has not yet been proven in ME. This study evaluated the potential of the histone methylation inhibitor chaetocin and a CARM1 inhibitor, as well as the histone phosphorylation inhibitors aurora kinase inhibitor II (AUKI-II) and hesperadin, to improve the efficiency of the early stages of ME induction in two distinct bread wheat cultivars with different ME responses. The effects of these epigenetic modifiers were compared with those of TSA. These compounds were applied simultaneously with a mannitol stress treatment, as this was the most effective strategy for TSA application in wheat [21]. Previous studies have shown that the application of DMSO during the stress treatment increased the final ME efficiency, and DMSO exerted an additive effect with TSA in wheat [21,31]. Therefore, the results for each modifier were compared, not only with the control, but also with cultures treated with DMSO, to evaluate the effect of the compound itself.

### 3.1. Effect of Different Epigenetic Modifiers on the Early Stages of Wheat ME Induction

The effect of epigenetic modifiers was determined by quantifying the percentages of the most representative structures present during the first 10 days in culture. This approach has been revealed to be effective in identifying ME inducers in different species [9].

For the first time, the ability of chaetocin, a CARM1 inhibitor, and AUKI-II to change the developmental pattern of microspores in relation to embryogenesis was demonstrated. However, it should be noted that the rate of embryogenic structures varied depending on the specific modifier and the cultivar (Figure 1B). Assessing to the higher ME response of the cultivar Pavon [21,32], a higher percentage of embryogenic structures was observed in this cultivar with all the epigenetic modifiers.

Chaetocin is an inhibitor of SU(VAR) 3-9 histone lysine methyltransferase, and is responsible for H3K9 methylation [25]. These enzymes play a key role in several plant growth and development processes, including gametophyte development, flowering, plant morphology, and stress response [23]. To the best of our knowledge, BIX-01294 is the only histone methyltransferase inhibitor that has been tested in ME, but this compound is not effective in wheat [20]. However, the application of chaetocin to tobacco seedlings for one month, in combination with the auxin 2,4-dichlorophenoxyacetic acid treatment, resulted in increased callus formation [33]. In our study, chaetocin produced a higher percentage of tetracellular embryogenic structures than the control, resulting in a 1.7-fold increase compared to TSA in Caramba. In Pavon, this compound also yielded a higher percentage of multicellular structures compared to the control, but this was slightly lower with respect to TSA. A previous study has already established differences between these cultivars in the presence of the H3K9me2 mark before and after induction treatment with mannitol [32]. The mark was absent in the high-responding cultivar Pavon, but was found to be present in the mid-low-responding cultivar Caramba.

CARM1/PRMT4 is an enzyme that catalyzes the arginine methylation of histones (H3R17me2 and H3R26me2) and other proteins [34]. In plants, arginine methyltransferases (PRMTs) form a conserved family involved in multiple essential cellular processes and the regulation of crucial traits [35,36]. In Pavon, the CARM1 inhibitor induced higher percentages of tetracellular and multicellular structures than the control, although the percentage was slightly lower than that of TSA (Figure 2A). Conversely, this compound produced one of the highest percentages of tricellular pollen-like structures, but a very low number of tetracellular structures, and none of the multicellular embryogenic variety, in Caramba (Figure 2B). All these data indicate that this epigenetic modifier was unable to change the microspore developmental pattern in the mid-low-responding cultivar.

Plant AUK can phosphorylate histone H3 and microtubule binding proteins such as MAP65-1, among other proteins [37]. In *Arabidopsis*, aurora kinases regulate the response to abiotic or biotic stimuli, modulating developmental processes and signal transduction alike [38]. Plant β-Aurora kinases exert a key role in cell division by regulating centromere function [26,27,39]. The application of hesperadin, an inhibitor of β-aurora kinase, leads to chromosome segregation disturbances in BY-2 cells extracted from tobacco [29,30]. In Caramba, hesperadin produced a significantly higher number of tricellular pollen-like structures than the other epigenetic modifiers and the two controls. Additionally, hesperadin showed a slightly higher percentage of tetracellular structures than the control, but a similar level to the control DMSO in both cultivars. These results suggest that the inhibition of β-aurora kinases is not a robust strategy for enhancing ME in wheat (Figure 2).

α-aurora kinases play a fundamental role in the orientation of the asymmetric cell division plane during lateral and apical root meristem formation, affecting the early stages of embryo development, as well as pollen viability and vascular differentiation [26,28,40,41,42]. This study demonstrated that AUKI-II had the highest potential to induce ME in both cultivars based on the percentage of multicellular structures. The rate of embryogenesis with AUKI-II was slightly higher than that with TSA (Figure 2).

Thus, the screening of epigenetic modifiers performed in this study highlighted the potential of AUKI-II to induce embryogenic structures in two cultivars of wheat with different ME responses. Therefore, we evaluated the effect of AUKI-II on the final efficiency of ME.

### 3.2. Effect of Aurora Kinase Inhibitor II on the Efficiency of Green DH Plant Production

The number of green DH plants is largely influenced by the concentration, duration, and application time of an epigenetic modifier, as well as their interactions. This has been previously reported with TSA in *Brassica* and wheat [10,12,17,19,20,21]. In this study, AUKI-II was applied during stress treatment with mannitol, following two different approaches: a short treatment, as used in compound screening, and a long treatment, which was found to be the most effective method when previously used with TSA [21].

Although the application of 0.4 µM for 24 h in SM liquid medium produced the highest percentage of multicellular embryogenic structures, neither 0.4 µM nor 0.8 µM AUKI-II yielded a higher number of more advanced structures in development (pro-embryos) than any control in the case of Pavon (Table 1). Similarly, TSA or even DMSO had no effect on the number of pro-embryos compared to the control. Additionally, AUKI-II did not increase the number of embryos or green plants. These data suggest that a high percentage of multicellular embryogenic structures were unable to produce pro-embryos with this short treatment. Similar results have been reported in *Brassica* with high concentrations of TSA [10,43]. Our observations emphasize the importance of assessing the activity of ME inducers during n the early stages of culture and throughout plant development.

One of the primary issues with wheat ME is the low rate of spontaneous chromosome doubling. This leads to the use of toxic compounds such as colchicine, orizalin, AMP, etc., which can result in the loss of DH lines and the need for an additional cycle of seed multiplication [44]. DMSO has been found to exert contradictory effects on chromosome doubling. Jakše et al. [45] reported negative effects on onion embryos exposed to 2% DMSO, whereas Hooghvorst et al. [46] found that 1% DMSO did not affect rice seedlings. In this study, the use of DMSO decreased the percentage of spontaneous chromosome doubling. However, 0.4AUKI-II and 0.8AUKI-II counteracted the effect of DMSO in Pavon. These results agree with those of the study conducted by Demidov et al. [40] in *Arabidopsis*. They applied AUKI-II to the apical meristem of diploid plants, resulting in tetraploid and aneuploid plants.

Since a short application treatment with AUKI-II was not effective, we tested a longer treatment, which was the best strategy for TSA application [21]. AUKI-II was applied for 5 days in SM medium with 1% DMSO. In addition, a higher concentration of AUKI-II (1.2 µM) was used because, in the previous experiment, the highest concentration tested (0.8 µM) showed the most favorable results (Table 1). In Pavon, 1.2 µM AUKI-II increased the percentage of plant regeneration, and 0.8 µM AUKI-II slightly increased the percentage of chromosome doubling compared to the control. However, none of these concentrations had any effect on the final ME efficiency (Table 3). Conversely, in the mid-low-responding cultivar Caramba, 0.8 µM AUKI-II increased the percentage of doubling and the final ME efficiency compared to the control DMSO. Furthermore, 0.8AUKI-II produced 33% more green DH plants than the control due to a slight increase in different variables.

Compounds such as colchicine and n-butanol, which affect the stability of spindles and endoplasmic and cortical microtubules, have been shown to enhance ME efficiency by increasing the percentage of chromosome doubling [47,48,49]. α-aurora kinases phosphorylate microtubule-associated proteins, which are required for efficient cell cycle progression [37,50]. Furthermore, α-aurora kinases also play a role in cortical microtubule assembly in ferns [51]. Our results suggest that AUKI-II has the potential to induce spontaneous chromosome doubling in two bread wheat cultivars, which is probably due to its impact on microtubule assembly rather than being caused by histone modifications.

This study evaluated the potential of epigenetic modifiers acting on histone methylation (chaetocin and a CARM1 inhibitor) and histone phosphorylation (AUKI-II and hesperadin) to induce ME in wheat. These compounds have not been used previously for this purpose. Although the results we obtained with chaetocin seemed promising in terms of the percentage of tetracellular or multicellular embryogenic structures in both cultivars, its effect was not superior to that of TSA. However, AUKI-II showed the highest potential, with slightly better percentages than TSA. Our results suggest that AUKI-II has the potential to induce spontaneous chromosome doubling in two bread wheat cultivars. The application of 0.8 µM AUKI-II during a long stress treatment did not alter the final ME efficiency in a high-responding cultivar, but instead produced 33% more green DH plants than the control in the mid-low-responding cultivar. The effect of AUKI-II should be assayed in relation to a wider range of genotypes. Furthermore, new application strategies similar to those with colchicine or n-butanol could be effective. In addition, new epigenetic modifiers exhibit promise in terms of increasing the efficiency of other systems that require cellular reprogramming.

## 4. Materials and Methods

### 4.1. Growing Conditions of Donor Plants and Harvesting of Spikes

In this study, we used anthers from the bread wheat cultivars Pavon and Caramba, which have high and mid-low ME responses, respectively, as well as ovaries from Caramba.

Spike donor plants were grown according to the procedure described by Castillo et al. [52]. For preparation of the anther culture and ovary pre-conditioned medium (OVPCM), excised spikes were sprayed with 96% ethanol for sterilization. Anthers containing microspores at the mid-to-late uninucleate stage, as determined by DAPI (4′,6-diamidine-2′-phenylindole dihydrochloride) staining, were used for anther culture. For OVPCM, mature ovaries from flowers containing late binucleate microspores were cultured [52].

### 4.2. Anther Stress Pretreatment

Anthers were inoculated in stress medium (SM), which contained 0.7 M mannitol and 5.9 g/L CaCl_2_ 2H_2_O plus the macronutrients from the FHG medium [53] and 8 g/L agarose, for 5 days at 25 °C [53], unless otherwise stated.

### 4.3. Preparation of Ovary Pre-Conditioned Medium

Six mature ovaries from Caramba were cultured in a 3 cm Ø Petri dish with 2 mL of MSMIF2 medium [53], which is a modified MMS3 medium [54] containing 1 mg/L of 2,4-dichlorophenoxyacetic acid (2,4-D) and 1 mg/L of benzyladenine (BA) supplemented with Ficoll type 400 (200 g/L). The cultures were kept in the dark at 25 °C for 5 days before undergoing anther culture. To ensure uniformity between the treatments, ovaries from the same spike were randomly distributed among treatments from the same replicate. Ovaries were maintained in the medium throughout the anther culture [53].

### 4.4. Anther Culture

Anthers were cultured in MSMIF2 ovary pre-conditioned medium in the dark at 25 °C. After 10–12 days, the cultures were replenished with 2 mL of MSMIF4 containing 400 g/L of Ficoll type 400 [54]. After 40 and 60 days, the mature developed embryos were transferred to the regeneration medium J25-8 [55]. Embryos were maintained in the dark at 25 °C for 2 days and then transferred to light. 

### 4.5. Screening of Different Compounds to Induce ME in Wheat

The epigenetic modifiers were applied to SM liquid medium (without agarose) (control, L-CM) for 24 h at 25 °C. Two histone methylation inhibitors, HMTase Inhibitor II Chaetocin (Merck KGaA, Darmstadt, Germany) and a CARM1 inhibitor (Merck KGaA, Darmstadt, Germany), and two histone phosphorylation inhibitors, aurora kinase inhibitor II (AUKI-II) (Merck KGaA, Darmstadt, Germany) and hesperadin (Merck KGaA, Darmstadt, Germany), were assayed at 0.4 µM (L-0.4Chaetocin, L-0.4CARM1I, L-0.4AUKI-II, and L-0.4Hesperadin, respectively). A control was used to successfully induce ME with 0.4 µM TSA (Merck KGaA, Darmstadt, Germany) (L-0.4TSA). We implemented an additional control to differentiate the effect of the epigenetic modifiers from that of DMSO (L-CM + DMSO), since all epigenetic modifiers were dissolved in DMSO (final concentration in the medium 1%) [21]. Freshly excised anthers from the same spike were randomly distributed in 2 mL of the seven stress media mentioned above. In total, 7 spikes were used per replicate and at each time of sampling. Four replicates were performed, each with 40 anthers.

After the treatment, the anthers were transferred to MSMIF2 ovary pre-conditioned medium. Anthers were collected after 2, 4, and 10 days of culture (2dc, 4dC, and 10dC, respectively) and fixed in 4% paraformaldehyde in PBS with pH 7.3 for 24 h at 4 °C. After three washes in PBS for 10 min each, the anthers were kept in 0.1% paraformaldehyde in PBS with a pH 7.3 until isolation. Anthers from the 4 replicates were combined and transferred to an Eppendorf tube containing 600 mL of PBS for homogenization using a sterilized swab. The homogenate was filtered through a 100 µm nylon mesh and centrifuged at 3000 rpm for 1 min. The pellet was suspended in 150 µL of PBS. Microspores were counted using a hemocytometer and stained with DAPI, then excited at 350 nm in combination with a 400–420 nm long pass filter under an inverted Nikon Eclipse-T300 microscope.

### 4.6. Effect of Aurora Kinase Inhibitor II on the Efficiency of Green Plant Production

#### 4.6.1. Application of Aurora Kinase Inhibitor II during a 24-h Stress Treatment

AUKI-II at 0.4 or 0.8 µM and 0.4 µM TSA (L-0.4AUKI-II, L-0.8AUKI-II, and L-0.4TSA, respectively) were applied to SM liquid medium for 24 h at 25 °C. Control cultures in SM liquid medium (L-CM), as well as the control DMSO in SM liquid medium with 0.1% DMSO (L-CM + DMSO), were also included, since TSA and AUKI-II were dissolved in DMSO (final concentration in the medium: 0.1%). Freshly excised anthers from the same spike were randomly distributed in 2 mL of the five stress media mentioned above. After treatment, anthers were cultured in MSMIF2 ovary pre-conditioned medium.

#### 4.6.2. Application of Aurora Kinase Inhibitor II during a 5-Day Stress Treatment

AUKI-II at 0.4, 0.8 µM, or 1.2 µM (0.4AUKI-II, 0.8AUKI-II, or 1.2AUKI-II, respectively) was applied in SM medium for 5 days at 25 °C. Control cultures in SM medium (CM) and control DMSO in SM medium with 1% DMSO (CM + DMSO) were also included, since AUKI-II was dissolved in DMSO (final concentration in the medium: 1%). Freshly excised anthers from the same spike were randomly distributed in the five stress media mentioned above. Following 5 days of treatment, the anthers were cultured in MSMIF2 ovary pre-conditioned medium.

### 4.7. Stereoscopic and Microscopic Observation

The anther culture was visualized under an inverted Nikon Eclipse-T300 microscope and a SMZ750 Nikon binocular stereoscope (Nikon Europe B.V. Amstelveen, The Netherlands). The images of isolated microspores, pro-embryos, and embryos were captured using digital cameras (Digital sight DS-U2 and Digital sight DS-Fi1) and processed with NIS-Elements D.3.2 software (Nikon Metrology Europe N.V. Leuven, Belgium).

### 4.8. Ploidy Analysis

The ploidy level was estimated using flow cytometry with a PAS device (Partec-Sysmex). Small pieces of young leaves were chopped and filtered through a 30 µm nylon filter into 500 µL of Cystain UV ploidy solution (Partec-Sysmex). The nuclei were collected in a plastic tube compatible with the loading port of the cytometer. Two milliliters of MilliQ water were added and mixed thoroughly. The fluorescence peak of the control samples was fixed at X axis point 150 using leaves from young bread wheat seedlings. The ploidy level of the samples was inferred via comparison with the control. A minimum of 8000 counts per sample were recorded [52]. 

### 4.9. Statistical Analysis

To analyze the screening of the epigenetic modifiers, we counted the number of uninucleate microspores, bicellular structures after symmetric or asymmetric division, tricellular structures, pollen-like structures or embryogenic structures, and tetracellular or multicellular embryogenic structures enclosed inside the exine following 2, 4, and 10 days of culture. A total of 400–500 microspores from cv. Pavon and 250–300 microspores from cv Caramba were analyzed for each individual epigenetic modifier, taken from 5 different fields. The percentages of uninucleate microspores and structures with different numbers of cells were calculated over the total number of microspores present in each preparation. The FREQ procedure was used to perform the chi-squared test for analysis.

To evaluate the effect of AUKI-II on the efficiency of green DH plant production, we conducted 20–21 replicates per treatment using 30 and 36 anthers from Pavon and Caramba, respectively. Anthers from the 6 central flowers from each side of 5 spikes were randomly distributed among the five treatments within each replicate. The following variables were recorded: number of mature embryos (EMB: number of embryos with a well-developed embryo axis per 100 anthers), number of green plants (GP: number of green plants per 100 anthers), number of green DH plants (GPDH: number of green DH plants per 100 anthers), percentage of green plants (PGP: number of green plants per 100 total plants), percentage of regeneration (PREG: number of plants/100 embryos), and percentage of spontaneous chromosome doubling (PDH: number of DH plants per 100 plants). The number of pro-embryos (PEMB: number of multicellular structures outside the exine per 100 anthers) was calculated as the sum of the number of embryogenic calli that did not develop an embryo axis and the number of mature embryos. The number of embryogenic calli was estimated by counting the structures in one-tenth of the Petri dish area with a stereoscopic microscope and millimeter paper on day 60 of culture, after all mature embryos had been transferred for regeneration. 

Statistical analysis was performed using SPSS software (Version 27.0 IBM Corp). Normality and homogeneity of variance were tested using Kolmogorov–Smirnoff and Levene’s tests, respectively. The data were transformed using the square root (x + 0.5) to meet the parametric assumptions, except for the percentage of green plants, which did not require transformation. The GLM (generalized linear model) procedure was used to perform the ANOVA for all variables, except for the percentage of spontaneous chromosome doubling, which was instead analyzed via the use of the FREQ procedure to perform the chi-squared test. Duncan’s test (α ≤ 0.05) was used for means separation.

## Figures and Tables

**Figure 1 plants-13-00772-f001:**
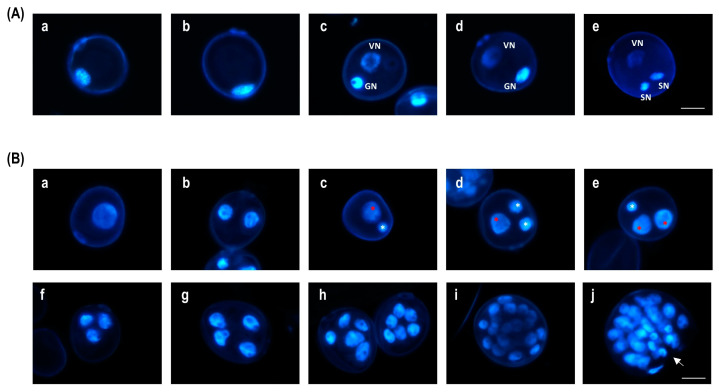
DAPI staining during in vivo pollen development and ME in bread wheat. (**A**) In vivo pollen development: microspore at medium uninucleate stage (**a**), microspore at late uninucleate stage (**b**), early bicellular pollen and late bicellular pollen with a generative and a vegetative cell ((**c**), (**d**) and (**e**), respectively), and mature pollen grain with a vegetative cell and two spermatic cells. (**B**) Most representative type of structures identified during the early stages of bread wheat anther culture: uninucleate microspore (**a**), bicellular structure after a symmetric division (**b**), bicellular structure after an asymmetric division (**c**), tricellular pollen-like structure (**d**), tricellular embryogenic structure (**e**,**f**), tetracellular embryogenic structure (**g**), and multicellular embryogenic structures enclosed inside the exine (**h**,**i**) and pro-embryo (multicellular structure with a broken exine) (**j**). GN: generative nucleus, VN: vegetative nucleus, SN: spermatic nucleus. Red asterisk: vegetative-like nucleus, white asterisk: generative-like nucleus, white arrow: exine rupture site. Scale bar = 20 µm.

**Figure 2 plants-13-00772-f002:**
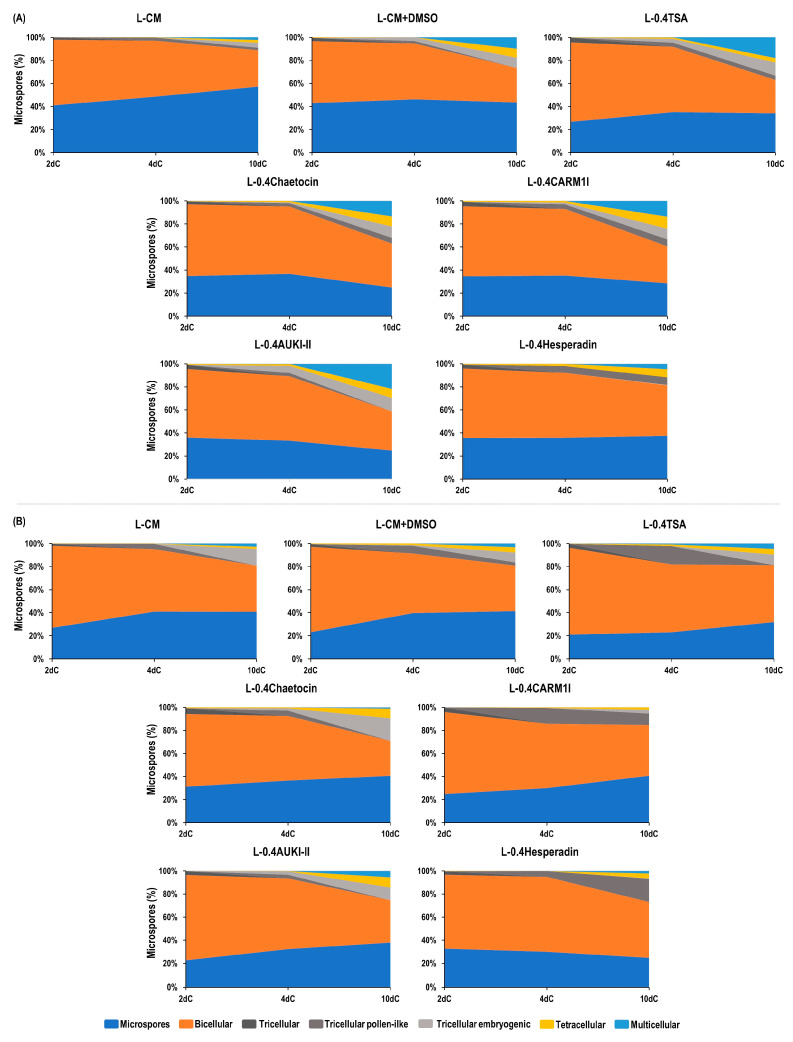
Effect of 24 h stress treatment in SM liquid medium with 0.4 µM TSA (L-0.4TSA), chaetocin (L-0.4Chaetocin), a CARM1 inhibitor (L-CARM1I), aurora kinase inhibitor II (L-0.4AUKI-II), and hesperadin (L-0.4Hesperdin) on the percentages of microspores, bicellular structures, tricellular structures (both pollen-like and tricellular), and tetracellular and multicellular embryogenic structures after 2, 4, and 10 days in culture (2dC, 4dC, and 10dC, respectively) in wheat cultivars (**A**) Pavon and (**B**) Caramba. L-CM = control in SM liquid medium; L-CM + DMSO = control DMSO in SM liquid medium with 1% DMSO.

**Figure 3 plants-13-00772-f003:**
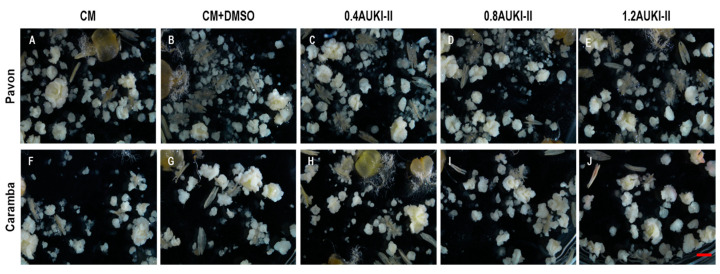
Effect of the application of 0.4, 0.8, and 1.2 µM AUKI-II (0.4AUKI-II, 0.8AUKI-II, and 1.2AUKI-II, respectively) during a 5-day stress treatment on the number of pro-embryos and embryos produced in Pavon anther culture (**A**–**E**) and Caramba (**F**–**I**) cultivars. (**A**,**F**) Control (CM) = SM medium; (**B**,**G**) control DMSO (CM + DMSO) = SM medium with 1% DMSO; (**C**–**E**,**H**–**J**) 0.4AUKI-II, 0.8AUKI-II, and 1.2AUKI-II = SM medium with 0.4, 0.8, or 1.2 µM of AUKI-II, respectively. Scale bar = 2 mm.

**Figure 4 plants-13-00772-f004:**
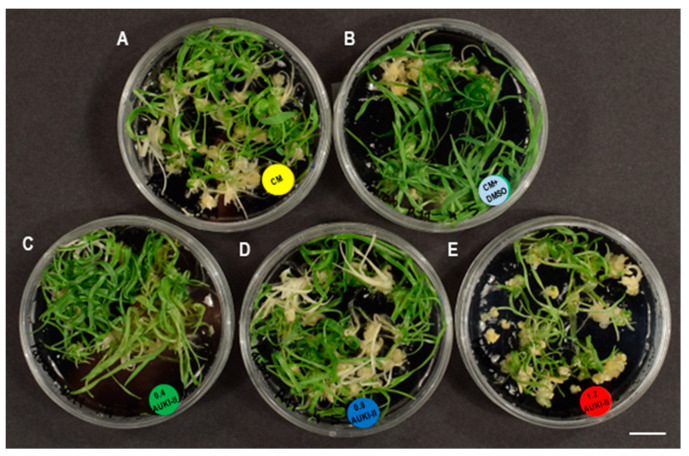
Effect of the application of 0.4, 0.8, and 1.2 µM AUKI-II (0.4AUKI-II, 0.8AUKI-II, and 1.2AUKI-II, respectively) during a 5-day stress treatment on Pavon plant regeneration. (**A**) Control (CM) = SM medium; (**B**) control DMSO (CM + DMSO) = SM medium with 1% DMSO; (**C**–**E**) 0.4AUKI-II, 0.8AUKI-II, and 1.2AUKI-II = SM medium with 0.4, 0.8, or 1.2 µM AUKI-II. Scale bar = 1 cm.

**Table 1 plants-13-00772-t001:** Effect of the application of 0.4 and 0.8 µM AUKI-II (L-0.4AUKI-II and L-0.8AUKI-II, respectively), and 0.4 µM TSA (L-0.4TSA) during a 24 h stress treatment in SM liquid medium on the final ME efficiency of Pavon wheat cultivar. L-CM = control in SM liquid medium. L-CM + DMSO = control DMSO in SM liquid medium with 0.1% DMSO. The table shows treatment means after analysis of variance for the number of pro-embryos (PEMB), embryos (EMB), green plants (GP), albino plants (AP), and spontaneously green DH plants (GPDH) calculated per 100 anthers, as well as percentages of green plants (PGP, number green plants/total plants), plant regeneration (PREG, number of plants/100 embryos), and spontaneous chromosome doubling (PDH, number of DH plants/100 plants).

Treatment	PEMB *	EMB *	GP *	AP *	PGP *	PREG *	PDH **	GPDH *
L-CM	482.9 a	135.6 a	17.0 a	27.6 a	36.8 a	29.0 b	65.8 a	11.2 a
L-CM + DMSO	434.4 a	123.7 a	26.2 a	28.4 a	44.3 a	47.3 a	46.4 b	12.2 a
L-0.4AUKI-II	376.7 a	101.3 a	15.6 a	18.7 a	47.3 a	35.5 ab	59.1 a	9.3 a
L-0.8AUKI-II	418.7 a	123.1 a	22.1 a	26.9 a	45.3 a	38.8 ab	68.5 a	15.2 a
L-0.4TSA	370.6 a	112.4 a	20.9 a	22.2 a	48.8 a	28.2 ab	45.4 b	9.5 a

Values expressed per 100 anthers. Values followed by the same letter within each variable are not significantly different (*p* < 0.05) according to a Duncan (*) or a chi-squared (**) test.

**Table 2 plants-13-00772-t002:** Analysis of variance of the effect of the application of 0.4, 0.8, and 1.2 µM AUKI-II (0.4AUKI-II, 0.8AUKI-II, and 1.2AUKI-II, respectively) during a 5-day stress treatment in SM medium on the final ME efficiency in Pavon and Caramba wheat cultivars. Control in SM medium (CM) and control DMSO with 1% DMSO (CM + DMSO) were also included. The variables of number of pro-embryos (PEMB), embryos (EMB), green plants (GP), and albino plants (AP) were calculated per 100 anthers. Percentages of green plants (PGP: number of green plants/100 total plants) and plant regeneration (PREG: number of plants/100 embryos) were also determined.

*p*-Value *	PEMB	EMB	GP	AP	PGP	PREG
Genotype (G)	˂0.001	˂0.001	˂0.001	0.104	˂0.001	0.746
Treatment (T)	0.980	0.794	0.608	0.155	0.735	0.353
G × T	0.097	0.568	0.850	0.882	0.795	0.620

* *p*-values of the analysis of variance for log-transformed data for all the variables except for PGP.

**Table 3 plants-13-00772-t003:** Effect of the application of 0.4, 0.8, and 1.2 µM of AUKI-II (0.4AUKI-II, 0.8AUKI-II, and 1.2AUKI-II, respectively) during a 5-day stress treatment in SM medium on the ME efficiency of Pavon and Caramba wheat cultivars. CM = control culture in SM medium. CM + DMSO = control DMSO in SM medium with 1% DMSO. Table 3 shows the treatment means after analysis of variance for the number of pro-embryos (PEMB), embryos (EMB), green plants (GP), albino plants (AP), and green DH plants (GDH), calculated per 100 anthers, and the percentages of green plants (PGP: number green plants/100 total plants), plant regeneration (PREG: number of plants/100 embryos), and spontaneous chromosome doubling (PDH: number of DH plant/100 plants).

Treatments	PEMB *	EMB *	GP *	AP *	PGP *	PREG *	PDH **	GPDH *
Pavon								
CM	767.0 a	116.8 a	40.2 a	19.2 a	64.4 a	51.0 ab	57.3 a	23.0 a
CM + DMSO	668.1 ab	115.8 a	36.4 a	14.1 a	72.4 a	42.4 b	56.0 a	20.4 a
0.4AUKI-II	647.3 b	110.0 a	33.8 a	13.0 a	73.5 a	41.7 b	61.5 a	20.8 a
0.8AUKI-II	663.8 ab	113.2 a	33.2 a	13.7 a	71.2 a	43.0 b	64.5 a	21.4 a
1.2AUKI-II	603.0 b	105.9 a	37.8 a	16.2 a	71.0 a	52.6 a	59.8 a	22.6 a
Caramba								
CM	282.3 a	54.3 a	8.5 a	14.3 a	37.7 a	54.2 a	61.7 a	4.6 ab
CM + DMSO	352.8 a	50.5 a	7.8 a	12.3 a	43.4 a	48.3 a	44.3 bc	3.1 b
0.4AUKI-II	373.2 a	42.8 a	7.3 a	10.3 a	34.9 a	47.4 a	39.0 c	2.6 b
0.8AUKI-II	372.8 a	57.7 a	11.0 a	13.2 a	38.0 a	47.1 a	63.1 a	6.1 a
1.2AUKI-II	393.0 a	63.8 a	10.2 a	12.5 a	41.9 a	47.2 a	58.0 ab	5.2 ab

Values expressed per 100 anthers. Values followed by the same letter within each variable are not significantly different (*p* < 0.05), according to the Duncan (*) or chi-squared (**) tests.

## Data Availability

Data are contained within the article and Appendix A.

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
