# Peer review of "New Epigenetic Modifier Inhibitors Enhance Microspore Embryogenesis in Bread Wheat"

_plants, 2024, doi:10.3390/plants13060772_

Round 1

Reviewer 1 Report

Comments and Suggestions for Authors

Valero-Ribira and co-authors describe the effect of a number of chemicals that inhibit epigenetic modifiers on microspore embryogenesis in two wheat cultivars. While I appreciate that the experiments were technically well-performed, I have serious concerns about their interpretation of the data. In general, their conclusions are either not supported by the data or overstated.

Figure 1, Figure 2 and Supplemental Tables 1 and 2

Main comments:

Figure 1, Figure 2 and the accompanying Supplemental Tables 1 and 2 present the most interesting data in the paper (possible early-stage cell proliferation induced by DMSO/compound treatments, but are difficult to interpret without additional information. The authors count the proportion of structures with one to  four or more nuclei after different treatments in the two genotypes but it is unclear whether these are pollen-like structures or embryogenic/vegetative structures or a mix of both. Moreover, no statistical analysis was performed, thus the significance of the data is not clear. I assume that this is a representative experiment rather than a one-off experiment i.e. that additional biological replicates show the same trend. This additional data should be included in the manuscript. Therefore, although the data are potentially interesting, the lack of careful description and statistical analysis make it impossible to determine the effect of DMSO/the epigenetic modifying compounds on the progression of microspore embryogenesis.

Minor comments:

The authors describe pollen-like structures, symmetrical/asymmetrical divisions, and nuclear condensations states, but these observations are not highlighted with marks in Figure 2. This makes it difficult for a non-expert to interpret the data.

The authors refer to uninucleate or binucleate microspores (line 99), but a microspore is, by definition, uninucleate. Therefore, the multicellular ‘microspores’ they describe are either pollen-like or embryogenic/vegetative structures.

Tables S1/2, ‘tetranucleate’ is not correctly spelled.

Table 1 and Table 2

Main comments:

I have serious concerns about the interpretation of the data in these tables. A quick look at the statistical groups assigned to the data show that for most parameters there is no statistically significant differences between treatments. When positive effects are assigned to a compound treatment, the compound treatment is often compared to the DMSO solvent control, which itself has a negative effect on the parameter compared to the DMSO-free control. If the goal is to find new/better compounds that enhance microspore embryogenesis, as implied in the Introduction, then such a compound is useless i.e. one could obtain more embryos in a culture that was not exposed to DMSO than in e.g. an AUKI-II-treated culture. It is misleading that the authors interpret their data in this manner.

Minor comments:

The rationale for choosing these compounds, among the thousands of compounds that affect epigenetic modifiers is not clear. If the rationale for this choice is based on unpublished data (line 57), then why not include this data in the manuscript?

It would be easier to read Tables 1 and 2 if the abbreviations (PEMB, EMB etc) were first defined in the text.

What is the difference between PEMB and EMB in terms of development?

GPDH is defined as GDH in the tables.

Comments on the Quality of English Language

The English language of the article could be improved. There are many instances when the subject of a sentence is not clear, making interpretation of the sentence difficult. For example, in lines 392-394, the sentence suggests that AUKI-II was added together with TSA ("it was not effective with TSA" should be replaced with "the same strategy was not effective with TSA").

Author Response

 Thank you for your suggestion. Please, see the enclosed file.

Reviewer 2 Report

Comments and Suggestions for Authors

Comments on the Quality of English Language

Author Response

Thnk you for your comments and suggestions. Please, see enclose file.

Reviewer 3 Report

Comments and Suggestions for Authors

In the study entitled “Aurora Kinase Inhibitor II: A Promising Compound for Micro-spore Embryogenesis in Bread Wheat”, authors have analyzed different inhibitors of epigenetic modifiers, such as Chaetocin, CARM1, Aurora kinase inhibitor II (AUKI-II), and Hesperadin to determine their potential to increase the number of pro-embryogenic structures. The study declares that AUKI-II was most effective, generating more pro-embryogenic structures. The study matter is interesting and of potential interest to the readers. However, the study involves several drawbacks, which need to be resolved and a major revision is suggested.

Please remove “.” from the title.

Figure quality needs to be improved.

Overall language of the manuscript needs to be improved.

Please make sure all the abbreviations are properly defined at the first mention.

Table 1: What does “GPDH” represent in table 1? Please define it in the legend of table 1.

Table 2: What does “PA” represent in table 2?

Line 33: wheat production must be increased by…

Lines 69-70: “Plant Aurora kinases (AUK) phosphorylate… key regulators of mitosis and meiosis.” Please re-write this sentence.

Line 72: Please remove “, according to Berenguer et al.”

Line 172: different concentrations of…

Line 314: responses showed differences in

Line 329: All these data indicate that…

Line 354: this inhibitor has the potential to cause

Line 358: ME responses.

Line 375: a high percentage was unable to…

Lines 413-414: It should be: “microtubule-associated proteins, such as AtTPX2 and MAP65-1 are required for efficient cell cycle progression”.

Line 426: “was higher in the mid-low responding one.” Please mention the name of the cultivar here.

Line 429: It should be Aurora kinase inhibitor II.

Lines 513-515: “Experiments to evaluate the effect of… Caramba, respectively.” Please re-write this sentence.

Comments on the Quality of English Language

Overall language of the manuscript needs to be improved.

Author Response

Thank you for your comments and suggestions. Please, see the enclosed file.

Round 2

Reviewer 1 Report

Comments and Suggestions for Authors

The revisions implemented by the authors have improved the readability of the manuscript. I have a few remianing comments:

1.  I disagree that a large number of buds from a single batch of plants grown at the same time constitutes a replicate. Microspore embryogenesis responses are highly variable, and small differences between treatments, as shown here, can disappear in a new batch of plants. To be practically applicable and mechanistically meaningful, culture treatments should produce a significant enhancement above the appropriate controls in different batches of plants/at different times of the year. Let's agree to disagree.

2. Figure 1. I appreciate the more careful description of the multicellular structures found in culture as either pollen or embryo-like. However, I feel that for non-experts it would be clearer to show the series of pollen development in planta, followed by panels with the types of development observed in vitro (as currently shown). Also, the generative and sperm cells should be labeled.

3. On line 111 the authors refer to a pollen with a vegetative nucleus and two generative nuclei. There is only one generative nucleus in wheat, that divides to form to sperm cells. So I think they mean two sperm nuclei not two generative nuclei.

4. Figure 2/Supplementary Tales 1 and 2: Why are the tricellular structures observed after 24 hours and 2 days of culture not separated into tricellular pollen-like and tricelluar embryogenic as in the later timepoints? As such, the term tricellular adds no extra information to the figure.

Comments on the Quality of English Language

Improved.

Author Response

Reviewer 1

Comments and Suggestions for Authors

Dear Review,

Thank you for your comments and suggestions regarding the manuscript.

The revisions implemented by the authors have improved the readability of the manuscript. I have a few remaining comments:

  1. I disagree that a large number of buds from a single batch of plants grown at the same time constitutes a replicate. Microspore embryogenesis responses are highly variable, and small differences between treatments, as shown here, can disappear in a new batch of plants. To be practically applicable and mechanistically meaningful, culture treatments should produce a significant enhancement above the appropriate controls in different batches of plants/at different times of the year. Let's agree to disagree.

We agree that there is significant variability between replicates/batches of plants in ME. It is also true that the screening experiment, as set up, cannot definitively conclude how the compounds act. More repetitions of the experiment would be necessary for this. However, even with increased repetitions, it is uncertain what the outcome would be if only the initial stages of the culture were analyzed. Other variables, such as embryo quality, percentage of albinism, or duplication, must be considered.

With our experiment we aimed to reach a compromise between identifying possible candidates, eliminating those with a negative effect on EM, and doing so quickly. Although the differences between AUKI-II and controls or TSA were minimal during screening, we studied their effects on a larger scale. However, we were aware that the results might differ.

  1. Figure 1. I appreciate the more careful description of the multicellular structures found in culture as either pollen or embryo-like. However, I feel that for non-experts it would be clearer to show the series of pollen development in planta, followed by panels with the types of development observed in vitro(as currently shown). Also, the generative and sperm cells should be labeled.

We have included a new Figure (Figure 1A) showing pollen development in vivo, and the vegetative, generative and sperm cells have been labelled.

  1. On line 111 the authors refer to a pollen with a vegetative nucleus and two generative nuclei. There is only one generative nucleus in wheat, that divides to form to sperm cells. So I think they mean two sperm nuclei not two generative nuclei.

This terminology has been corrected.

  1. Figure 2/Supplementary Tales 1 and 2: Why are the tricellular structures observed after 24 hours and 2 days of culture not separated into tricellular pollen-like and tricelluar embryogenic as in the later timepoints? As such, the term tricellular adds no extra information to the figure

We selected the compound with the greatest potential to induce ME based on data from 4 and 10 days in culture. On the other hand, the percentage of tricellular structures was very low at 2 days in culture, so we did not differentiate between the tricellular, pollen-like and embryogenic structures.

Reviewer 2 Report

Comments and Suggestions for Authors

MS is recommended for publication

Comments on the Quality of English Language

Author Response

No more comments.

Reviewer 3 Report

Comments and Suggestions for Authors

The manuscript, “Aurora Kinase Inhibitor II: A Promising Compound for Microspore Embryogenesis in Bread Wheat”, has been extensively revised. However, some minor changes regarding language and description are still required. The suggestions can be found below:

Please remove “to” from the title.

Some of the sentences are too long to understand.

Lines 88-92: “The RNA-seq analysis of…such as Aurore kinase” Please re-write this sentence.

Line 350: …present during the first 10 days in culture.

Comments on the Quality of English Language

The language is fine; however, some of the sentences are too long to understand. 

Author Response

Reviewer 3

Comments and Suggestions for Authors

The manuscript, “Aurora Kinase Inhibitor II: A Promising Compound for Microspore Embryogenesis in Bread Wheat”, has been extensively revised. However, some minor changes regarding language and description are still required. The suggestions can be found below:

 Dear Reviewer;

Thank you again for your comments on the manuscript.

Please remove “to” from the title.

This has been done.

Some of the sentences are too long to understand.

The English has been reviewed, with a special emphasis on this.

Lines 88-92: “The RNA-seq analysis of…such as Aurore kinase” Please re-write this sentence.

The sentence has been re-written.

Line 350: …present during the first 10 days in culture.

This has been done

Comments on the Quality of English Language

The language is fine; however, some of the sentences are too long to understand. 

The English Language Editing Services from MDPI have reviewed the text again, with a special emphasis on it.